sustainability

MDPI

Article

# Adoption-Driven Data Science for Transportation Planning: Methodology, Case Study, and Lessons Learned †

**Eduardo Graells-Garrido [1,2,*], Vanessa Peña-Araya [3] and Loreto Bravo [2]**

1 Barcelona Supercomputing Center (BSC), 08034 Barcelona, Spain
2 Data Science Institute, Faculty of Engineering, Universidad del Desarrollo, Santiago 7610658, Chile; bravo@udd.cl
3 LRI, CNRS, Inria, Université Paris-Saclay, 91190 Paris, France; vanessa.pena-araya@inria.fr
* Correspondence: eduardo.graells@bsc.es

† This paper is an extended version of our paper published in Eduardo Graells-Garrido and Vanessa Peña-Araya. 2020. Toward An Interdisciplinary Methodology to Solve New (Old) Transportation Problems. In Companion Proceedings of the Web Conference 2020 (WWW '20 Companion), Taipei, Taiwan, 20–24 April 2020.

Received: 29 June 2020; Accepted: 22 July 2020; Published: 26 July 2020

**Abstract:** The rising availability of digital traces provides a fertile ground for data-driven solutions to problems in cities. However, even though a massive data set analyzed with data science methods may provide a powerful and cost-effective solution to a problem, its adoption by relevant stakeholders is not guaranteed due to adoption barriers such as lack of interpretability and interoperability. In this context, this paper proposes a methodology toward bridging two disciplines, data science and transportation, to identify, understand, and solve transportation planning problems with data-driven solutions that are suitable for adoption by urban planners and policy makers. The methodology is defined by four steps where people from both disciplines go from algorithm and model definition to the development of a potentially adoptable solution with evaluated outputs. We describe how this methodology was applied to define a model to infer commuting trips with mode of transportation from mobile phone data, and we report the lessons learned during the process.

**Keywords:** transportation; urban mobility; data science; mobile phone data

## 1. Introduction

Cities are becoming increasingly complex, growing larger due to urbanization processes [1], and with increasing layers of interaction between their populations and their urban infrastructure [2]. The discipline of transportation is particularly affected by these issues, not only due to city growth but also because of the arrival of new technologies and changes in urban behavior. Another discipline, data science, has studied urban phenomena at previously unseen spatio-temporal granularity, mainly through the usage of mobile phone data [3]. Arguably, both disciplines complement each other: data science may provide tools to transportation to identify, understand, evaluate, and solve problems; transportation may provide important domain problems to be solved with data-driven approaches. However, both use different domain languages, resort to different data sources and methodologies, and have different priorities. Due to this gap between both disciplines, relevant stakeholders, such as public institutions and transportation authorities/operators, do not take advantage of the scalability, readiness, and granularity of data science-based models.

There is recognition on how a collaboration between these disciplines may deliver promising results [4,5]. Yet, the application of data-driven technologies might not solve the complex problems

of cities if they are not coupled with a range of other policies [6], making the collaboration between data scientists and policy makers crucial. Then, the question at hand is not the relevance of such collaboration—instead, it is how to conduct effective collaboration between them [7]. Such collaborations are needed, as a data-driven focus is not enough to evaluate the quality of a solution [8]. Indeed, the primary use of these modern data sources must "enable the creation of new knowledge by more people, not replace it" [4].

The relevance of collaboration between disciplines becomes clear when analyzing the actual situation of cities. In this ever-changing context, frameworks such as Data Collaboratives [9] allow one to define and kickstart interdisciplinary (transportation, data science) and cross-sector (public, private) initiatives that could bring societal value in a sustainable way for all stakeholders involved. Yet, questions that arise during project development, such as How to create such value? and How to ensure that the problem is solved in a way suitable for adoption? are outside of the framework's scope, as it focuses on establishing the collaboration and defining its value, rather than developing and executing the project. Moreover, even though data science generates either quantitative actionable knowledge, or data products [10], how to make those data products suitable for adoption in urban planning contexts is not clear. There are methodologies that put values and well-being as ultimate aims of a data science project [11], and while we agree that data science projects should strive for improving quality of life through social impact, this can be obtained only if its outcomes are adopted by relevant stakeholders. Indeed, there are extensive Big Data projects documented in the literature that cover transportation demand analysis from digital traces [12]. To the extent of our knowledge, it is not reported how or when these systems were put into operation in case they were adopted in real planning scenarios.

Most of the work about collaborative efforts between data science with other disciplines has focused on intermediary steps in the path toward adoption but not adoption itself. For instance, in the definition of liaison roles that improve the technical quality of a project [13], the definition of methods to synthesize the results of collaborative activities between stakeholders [14] and the definition of analysis approaches where domain experts have guided interactions with analytical systems to maximize their usefulness [15]. Here we propose to bridge the gap toward adoption through a four-step methodology that puts data science and transportation to work collaboratively to identify, understand, solve, and evaluate solutions to transportation planning problems. Some of these problems are not new, but the complexities of modern cities make the application of previously used methods unfeasible; at the same time, new methods may lack the necessary qualities to be adopted in planning and policy making [16].

In this work we describe each step of the methodology, including the stakeholders and concepts involved, as well as a case study of a method to infer the transportation mode share of the population in Santiago (Chile) using mobile phone data [17]. By guiding the project's development by the methodology, we obtained valuable insights for the development and adoption of data science solutions for urban problems. We report these insights and our reflections as lessons learned during the project execution. We believe that such insights will support broader collaboration of transport experts not only with data scientists but with citizens in general.

This paper is structured as follows. Section 2 describes the four-step methodology. Section 3 describes a case study for the problem of inferring the share of population that uses each mode of transportation in Santiago, Chile. Finally, Section 4 states our conclusions and next steps.

## 2. Methodology

Our main goal is to define a framework to guide the collaborative development of data-driven solutions for problems in transport planning. A transport planning problem is defined as a set of requirements to be fulfilled by urban transport planners and/or policy makers, which deals with issues regarding land use, transport supply, or traffic, and the intersections of these areas [18]. There is a long tradition of how to solve these problems using traditional methods and tools, however, there are

many challenges imposed by current needs in cities, derived from population growth, technological advances, the climate crisis, and other disruptions that change priorities. Nontraditional sources allow planners to work at finer time scales and analyze disruptive events [19]. These modern data sources may require advanced computational methods to build a data-driven solution, and thus, the need to follow a data science approach in designing and implementing it [10]. This raises several challenges because a research system that aims at having impact on public policy needs to be transparent and interpretable for policy makers, qualities that are not always a priority in these types of projects [16,20].

This methodology is designed to be applied after a transport planning problem has been identified. As such, it assumes that there are several stakeholders involved with the development and adoption of the potential solution. From the transportation domain, there are planners and practitioners that provide information about the problem to be solved and feedback for the proposed solution to be adopted. In addition, these stakeholders provide access to domain reference data to be used for evaluating the solution's performance. The data scientists from the data science's side provide the know-how to develop a solution. In order to bridge the communication between both disciplines, it is important for data scientists to also have knowledge on how to design effective visual representations of the solution. If it is not the case, then it would be necessary to include information visualization experts to contribute in this domain. Finally, there are the providers of the nontraditional data to be used for building the solution who might not necessarily belong to any of the involved disciplines but are equally relevant as stakeholders, as defined by the data collaboratives framework [9].

The methodology is summarized in Figure 1 and consists of the following steps:

1. **Algorithm Design:** to process both domain requirements and nontraditional datasets to reach a first version of the solution.
2. **Data-Driven Evaluation:** to evaluate the solution's performance in comparison to a domain reference dataset to be used for iterating the solution.
3. **Interactive and Collaborative Evaluation:** to evaluate if the solution is useful enough to be adopted in planning and policy making. Similar to the previous step, its output is used to iterate the solution.
4. **Knowledge Consolidation:** to congregate the generated knowledge and systems, with stakeholders assessing the requirements for the solution to be adopted in the real-world scenarios. Similar to the previous steps, its output is used to iterate the solution.

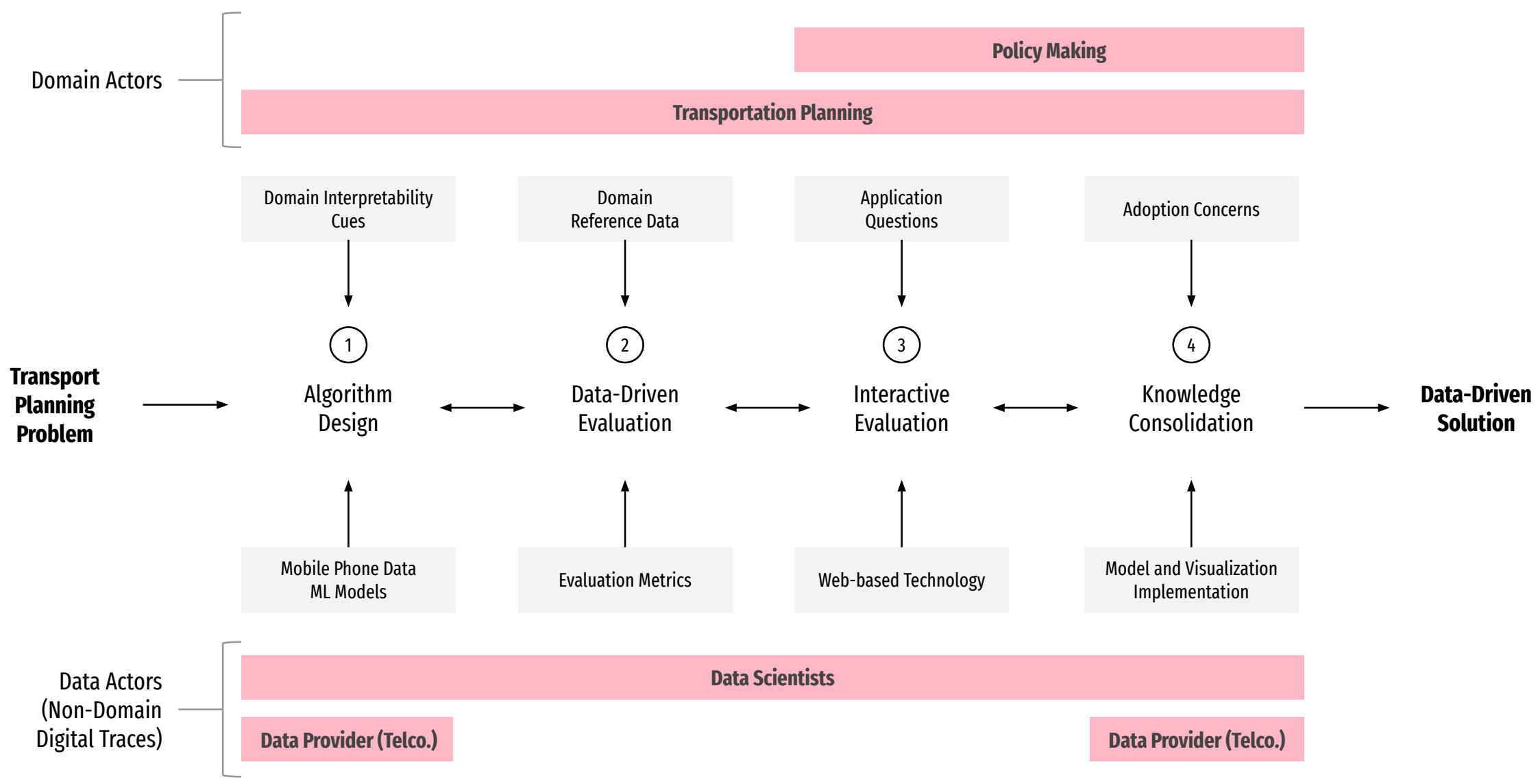


**Figure 1.** The four steps of the proposed methodology, with the relevant aspects contributed to each step from the several stakeholders involved, including data scientists, data providers, domain experts, and policy makers.

These four steps are sequential in appearance; however, the framework could be iterative or nested, depending on the development strategy followed in each project. Each step receives specific input and tools from each discipline, and all steps use tools from user research methods to enhance communication between people from different backgrounds [21]. This is important, for instance, to ensure that the technical language differences between disciplines converge into opportunities rather than limitations during the development of the solution. The final solution should consist in an evaluated data science model that addresses the requirements of the transportation problem and a set of visual representations of this model.

We describe each step in detail as follows.

### 2.1. Algorithm Design

The purpose of this step is to develop a model or algorithm that solves the problem in analytical terms. Traditionally, each discipline has their own processes to solve a problem through model development and algorithm design. Since we aim at creating a data-driven model to be incorporated in the transportation planning toolbox, we identify the contributions from each discipline into the model. On the one hand, data science may contribute new data sources that contain proxy information to solve the problem, as well as machine learning models to find patterns and make predictions. In contrast to traditional sources, the proxy information may not explicitly refer to the phenomena under analysis. For instance, a trip and its attributes (such as purpose and mode of transportation) may not be directly identifiable in a trajectory derived from mobile phone data. On the other hand, transportation contributes traditional data sources, domain knowledge, domain language, domain priorities, and interpretability cues, i.e., aspects of the problem and its potential solution that cannot be hidden in a black-box method.

From the model point of view, the development in this step is similar to a traditional data science process [10]. A key difference in our methodology is the incorporation of interpretability into the equation [16]. The difficulty here is the lack of consensus on what its definitions are. Two examples include “the degree to which an observer can understand the cause of a decision” [22] and “a method is interpretable if a user can correctly and efficiently predict the method’s result” [23]. Furthermore, different roles also have different interpretations [24]. Rather than delimiting what interpretability is, we propose to reach an agreement on its meaning for all project stakeholders. Then, a set of crucial questions in this stage of the project is: Taking into account the interpretability cues, how to converge definitions and priorities between disciplines? How to effectively define which pieces of the puzzle are going to be contributed from each discipline?

To answer those questions, we propose to execute several ideation instances while developing the model. We do not define how to conduct these codesign instances, as there are many alternatives, and the effectiveness of each one depends on the context. One interesting approach is the dialogue-labs method, which aims specifically at sparking dialogue between stakeholders [25]. Once the dialogue has allowed agreement, and the development has followed that dialogue, the output of this step is an algorithm or a model that has the potential to solve the problem at hand, and that may have the qualities of being interpretable by the stakeholders. Both aspects are evaluated in the next step.

### 2.2. Data-Driven Evaluation

Once there is agreement on the model components, a common evaluation in data-driven projects is to compare its performance with a reference data set, using metrics such as accuracy, precision, and recall, among others. However, as researchers have pointed out, lacking critical engagement with users can entail the failure of a machine learning system [11,26]. Therefore, the selection of accuracy metrics to contrast a ground truth data should be generated in collaboration with domain experts, and data scientists need to provide clear ways to explain these metrics to them. This is hard for two reasons. First, the availability of ground-truth data is not guaranteed. Domain experts may have data sources that describe urban phenomena, such as travel surveys, however, such sources may be

outdated or may be sparser than the data set available for the project, such as digital traces from mobile phone data. Second, even though the model components were designed to be interpretable, there is no guarantee that the model itself is transparent, i.e., operates in such a way that it is easy for stakeholders to see what actions are performed inside of it. This imposes a challenge because typical evaluation metrics do not necessarily measure the transparency of a model [20]. These challenges motivate the following questions to be answered at this point of development: *How does the model behave with validation data? What is the extent of the validation data? What are the intrinsic differences between the validation data and the nontraditional data? How does each interpretable feature from the model explain the phenomena?*

In this stage, we propose to codesign the evaluation criteria of the proposed model between data scientists and domain experts. This evaluation criteria must find the relationship between the model's explanation and its transparency, as well as balancing the trade-off between transparency and accuracy metrics. In order to assess the model in a way that allows all stakeholders to communicate, we propose to use visual representations as they allow to surface internal representations from users, helping to bridge potential gaps in analysis [27]. Since the design and evaluation of these visualizations is inherently a social process [28], this codesign process would not only generate artifacts that help to move the project forward but also formalize the common language between disciplines that started to form in the previous stage. Indeed, visualization is a tool valued by transportation researchers [29], and transportation has been a recurring topic in visualization [30].

The output of this step is a model that has been validated in its qualities of performance and transparency. Additionally, the output includes the first prototypes of visualizations of the model.

### 2.3. Interactive and Collaborative Evaluation

From the potential solution derived from the previous step, we still need to answer the following question: *Would domain experts use the solution in an applied context?*

For such validation, we propose to gather all stakeholders in a collaborative environment using large displays. Collaborative environments can facilitate interdisciplinary communication and development of shared ideas. In the context of data analysis, information visualization and big displays are powerful tools to enhance such environments. Together they put together a space where stakeholders engage with data science experts, providing feedback, asking new questions, framing results—enabling a continuous process of analysis [31,32]. Taking advantage of such an environment requires the design and implementation of technology, for instance, web-based visualization frameworks and the design of user research instruments to faithfully capture the interactions between experts [21].

Because adoption is our ultimate aim with the methodology, this stage should be strongly based on Human-Computer Interaction evaluation methods that ensure that the solution is appropriate for final users. The literature about how to design and evaluate systems from these two research areas is abundant, from which the works from Lam et al. [33] and from Isenberg et al. [34] provide good summaries. The adequate methods for each situation will depend on how the interactions in the collaborative environment are designed. If these methods are followed, then the output of this step is a model that has been validated in theoretical and applied contexts.

### 2.4. Knowledge Consolidation

In this stage stakeholders need to assess what is needed for the proposed solution to be adopted in a real planning scenario. We advance in that direction by jointly identifying concerns relevant for the adoption of the project. From their position, stakeholders look into dimensions that are orthogonal to the technical aspects covered in the previous stages. These include: definition of intellectual ownership and exploitation agreements with the data provider; the cost of accessing, storing, and updating the data; the compliance with privacy regulations; and the limits of population representativeness in the available digital traces, among others. Some of these concerns can be addressed by iterating over the

previous steps, while others require actions outside of the scope of this methodology, such as political agreements between private and public actors [9]. Others have been present from the beginning of the project; for instance, it is likely that the data provider and the research center may have an intellectual property agreement. A critical point of agreement is reached in this stage, when it is determined which part of the developed solution is apt for intellectual property protection (if any).

The need to consolidate the generated knowledge will differ according to the role of each stakeholder in the project. We describe two types of artifacts next.

On the one hand, there will be communication devices built considering insights and technical aspects of the solution, with information gathered through user research and communication practices. The audience of this device are the domain stakeholders. Its content is the consolidated knowledge of the project, aiming to be a useful tool to push for adoption by decision makers.

On the other hand, there will be a prototype data-product relevant for all stakeholders. Note that every component of the product does not need to be novel; the main challenge in this context is to integrate them into a novel solution for a real problem [35]. This product typically follows the dashboard or spatial visualization platform approach [36]. It is defined as a prototype and not as a final solution because there are issues regarding software development that are not covered by this methodology, as they fall out of its scope.

It is possible to claim that there is a potential data-driven solution for the initial transport planning problem once the development of these artifacts is completed.

In summary, this methodology provides guidelines on how to iteratively design and validate an interdisciplinary solution to a transportation planning problem. We have focused on identifying relevant questions that should be asked during the project, rather than a strictly defined list of inputs and outputs. This allows the methodology to be flexible for all stakeholders involved.

## 3. Lessons Learned from a Case Study

In this section we describe how the methodology was applied to the problem of inferring the mode of transportation share in Santiago (Chile) from anonymized mobile phone data [17,37].

Santiago has almost 8 million inhabitants, with an integrated multimodal transportation system. The city holds travel surveys every 10 years, a reasonable frequency rate in the previous decades, but this is inefficient in capturing the dynamics of the city today. Between the last surveys held, in 2012 and 2020, the city has transformed in unexpected ways—and not only in transportation terms. On the one hand, there are new ways of performing trips, such as ride-hailing applications [38], shared electric scooters, and ubiquitous routing services (e.g., Waze, an application that provides navigation and live traffic), which have changed how people make transportation choices. On the other hand, the population has acquired different habits. The last five years have seen an unprecedented international migration rate in the country, and migrants move differently through the city. Thus, the survey has its potential limits in planning and management of the transportation network, as the current transportation demand is unknown. Even though there is smart-card data that allows one to count the number of trips in public transport, fare evasion is another problem that affects the transportation system, with current rates above 25% that hide the actual public transport demand from planners [39].

Authorities and operators are actively searching for a new type of solution that allows them to complement the rich information of the travel survey with up-to-date observations of travel demand, not only to fulfill it, but also to improve the sustainability of the city. This situation prompted a collaborative effort between Telefónica Chile [40], the largest telecommunications company in the country, who provided data; the Data Science Institute in Universidad del Desarrollo [41], who provided know-how and research; and transportation practitioners from the Office of the Secretary of Transportation (SECTRA) in Chile [42], who provided domain knowledge and experience. There are several reasons that motivate the usage of mobile phone data by these institutions. First, it allows the data provider to leverage data already generated by its business, in a cost-effective procedure.

Second, it allows data scientists to perform applied research with a rich data source. Third, it allows transportation planners to understand how people move through the city at previously unseen granularities, both in terms of time and space.

This section is structured in the following way: for each step of the methodology defined in Section 2, we describe which stakeholders participated and how the work during that stage mapped to it. Note that the methodology was drafted initially; however, its current design benefited from the implementation of this case study, and it evolved in time until what it is defined in this document.

### 3.1. Algorithm Design

We worked with anonymized mobile phone data of type "Extended Detail Records" (XDR), consisting of billing records of data usage by smartphones. Each record contains the following metadata: *device identifier* (anonymous), *tower identifier*, and *timestamp*. The device identifiers are coherent in the dataset, which means that a spatio-temporal trajectory of users can be constructed from XDR. However, its usage in transportation analysis is not straightforward. For instance, towers have a connection coverage of around 1 kilometer on average. They are not distributed uniformly in the city, making the spatial granularity dense but not uniform. The temporal granularity is not uniform either, with varying frequencies during the day [43]. A trajectory can be constructed for each device, enabling the identification of trips [37], but care must be taken since this trajectory does not necessarily resemble a traditional route in the city (see Figure 2).

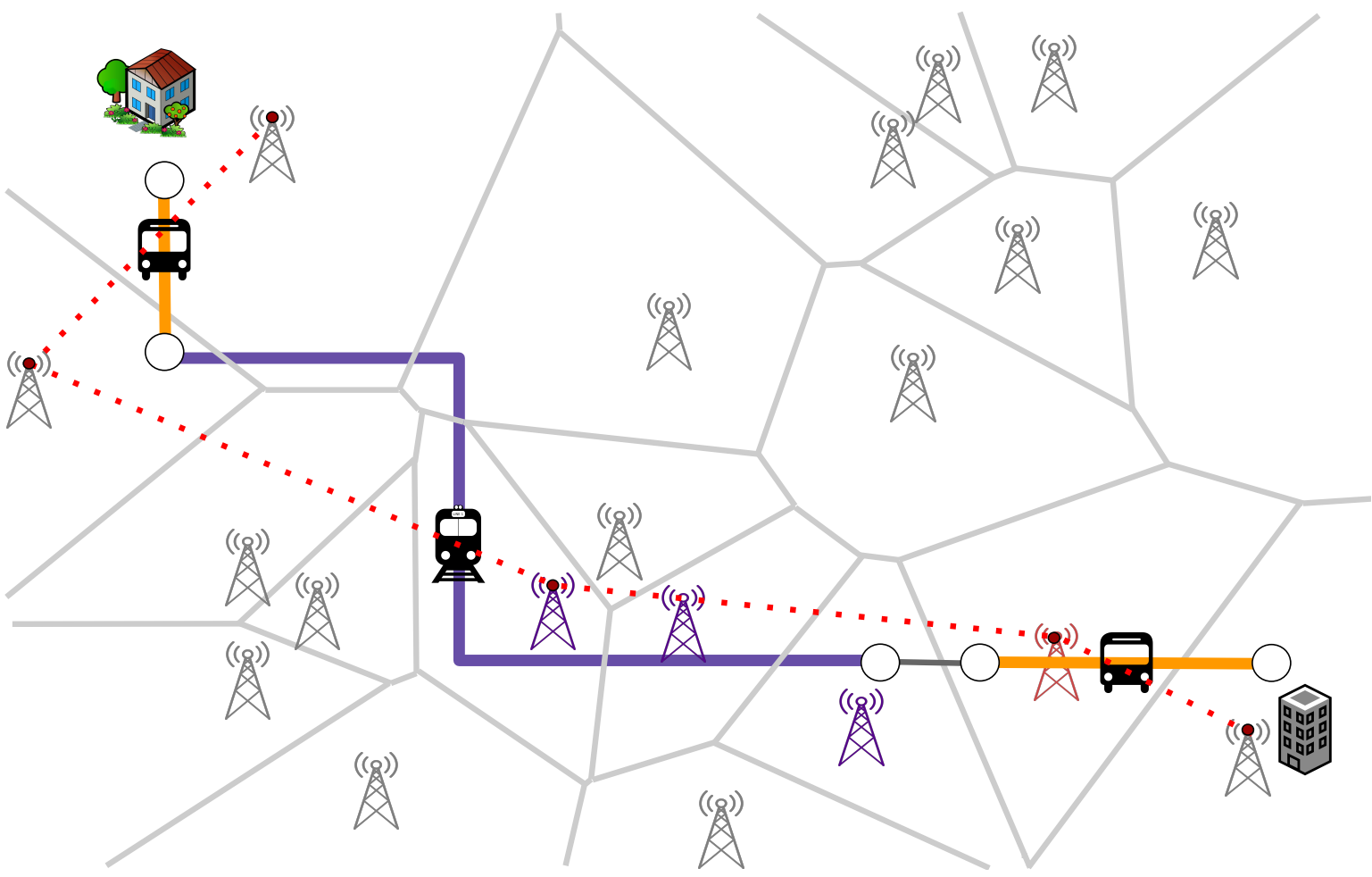

**Figure 2.** Representation of a commuting trip in mobile phone data. The sequence of towers connected to during a trip may have a small similarity with the real trajectory.

From a modeling perspective, we noted that single trajectories were not enough to reliably identify its mode(s) of transportation, which implied that we needed to aggregate user trajectories. Then, we identified that the situation could be described as a soft clustering problem, where each cluster is labeled with a mode of transportation. This rationale was based on how public transport users follow specific routes and how people tend to repeat their trajectories in commuting trips, which allowed us to aggregate them in terms of recurrent trips. After testing the accuracy of several clustering methods, we found that a model known as Topic-Supervised Non-Negative Matrix Factorization (TS-NMF) [44] was commonly assessed as an interpretable model due to is simple mathematical definition. Being a semisupervised model, it enables the integration of external knowledge in parts of the dataset. Other alternatives included different topic model and dimensional reduction methods.

Aiming to validate our assumptions, converging the model description to domain-language from both disciplines (data science and transportation), and integrating domain knowledge into the model, we organized two pilot workshops. One was held at a university and one at the premises of the telecommunication company. Attendees included planners from SECTRA, domain experts from

other universities, data scientists from the telecommunication company, and us. The format of the workshops was mainly based on presentations and question answering, which allowed plenty of discussion and opinion exchange. We learned that the model is not only interpretable because of its mathematical definition but also because the methods employed in finding a solution in TS-NMF are known in Transport Engineering, which commonly solves optimization problems, such as optimal bus fleet deployment. Regarding the integration of external knowledge, we agreed that feeding urban infrastructure information (such as distance from mobile phone towers to highways and bus lanes) improves the fine-tuning of the model, in addition to typical hyperparameter tuning.

We finished this stage with two outcomes. On the one hand, we developed a prototype implementation of the model, to be evaluated in the next stage. On the other hand, the open nature of our development process allowed us to create collaboration ties with other researchers and urban planners, which proved to be important in the future stages of the project.

### *3.2. Data-Driven Evaluation*

In this stage of the project, we aimed at assessing the performance and interpretability of the model in comparison to reference data. To do so, we hold several interview sessions with individual experts. As data scientists, our main concern was the absence of ground-truth data that would serve as a golden standard. Because the Santiago Travel Survey was from 2012 and our data was from 2016, we expected that they were not comparable. However, domain experts argued the comparison was still relevant and provided insight on which aspects they would evaluate in the model. Indeed, the modal share of the city has changed but not so drastically, and global mobility patterns are still fairly similar at the moment of the study.

In terms of performance, we learned that, since domain experts work with aggregated flows in the city, they were interested in the aggregated results of the model, i.e., origin and destination flows, both in general, and per mode, rather than in individual results. We agreed that a correlation coefficient between the origin-destination (OD) matrices would be enough for them to assess the quality of the model, taking into account that some OD pairs in the travel survey do not exist and should not be considered in the correlation. The general assessment would help them to validate the trip inference of our method, and the specific mode of transportation assessment would allow them to understand which mode(s) of transportation have changed their share at different spatial units.

The interviews revealed new knowledge too. We learned that our understanding of the term *interpretability* was different. For us (data scientists), it meant to have a clear image of the intermediate steps in the model, as a way to avoid black-box models. For them (domain experts), it was about the meaning of *error*, i.e., being able to identify where the model inference is wrong. In fact, a model with considerable error may be acceptable for experts, provided that this error is understood, and that measures to correct it can be taken. For instance, if the model incorrectly predicts mode of transportation for specific areas of the city, small, targeted surveys could be conducted, and such results would be apt to include in a new iteration of the model definition.

We expected that visualization could be a medium to strengthen the evaluation process by facilitating validation and interpretation. With domain experts, we codesigned a visualization to compare OD matrices (see Figure 3). The visualization shows two matrices side by side. The left matrix is row-normalized and depicts the number of commuting trips from/to all areas of the city, as inferred from mobile phone data. The right matrix shows the difference between the left matrix and an analogously built matrix from the travel survey. A divergent color palette in the second matrix helps to identify critical cells that can be analyzed manually by experts.

The design can be applied to all modes of transportation (see Figure 4), and it is flexible enough to include auxiliary plots. When displaying all modes of transportation, one can add a plot showcasing the residuals of the model flow inference, as it displays the distribution of error. The figure allows us to see, for instance, that there are three areas that attract metro trips from all over the city (middle column in the Figure). This was expected because these areas concentrate business and government districts.

However, what was not expected is how mobile phone data has more trips in metro toward these areas (second row, middle column): the three area columns are dominated by orange color, which, in the divergent color scale, means that more trips are observed in mobile phone data than in the travel survey. Note that metro trips are generally similar to the survey or with more observations in mobile phone data; conversely, when looking at the difference matrix in bus trips (first column), strong purple areas are more prominent, implying that less trips are observed in comparison to the survey. This is reflected on the correlations obtained for both modes of transportation (0.68 for metro and 0.62 for buses).

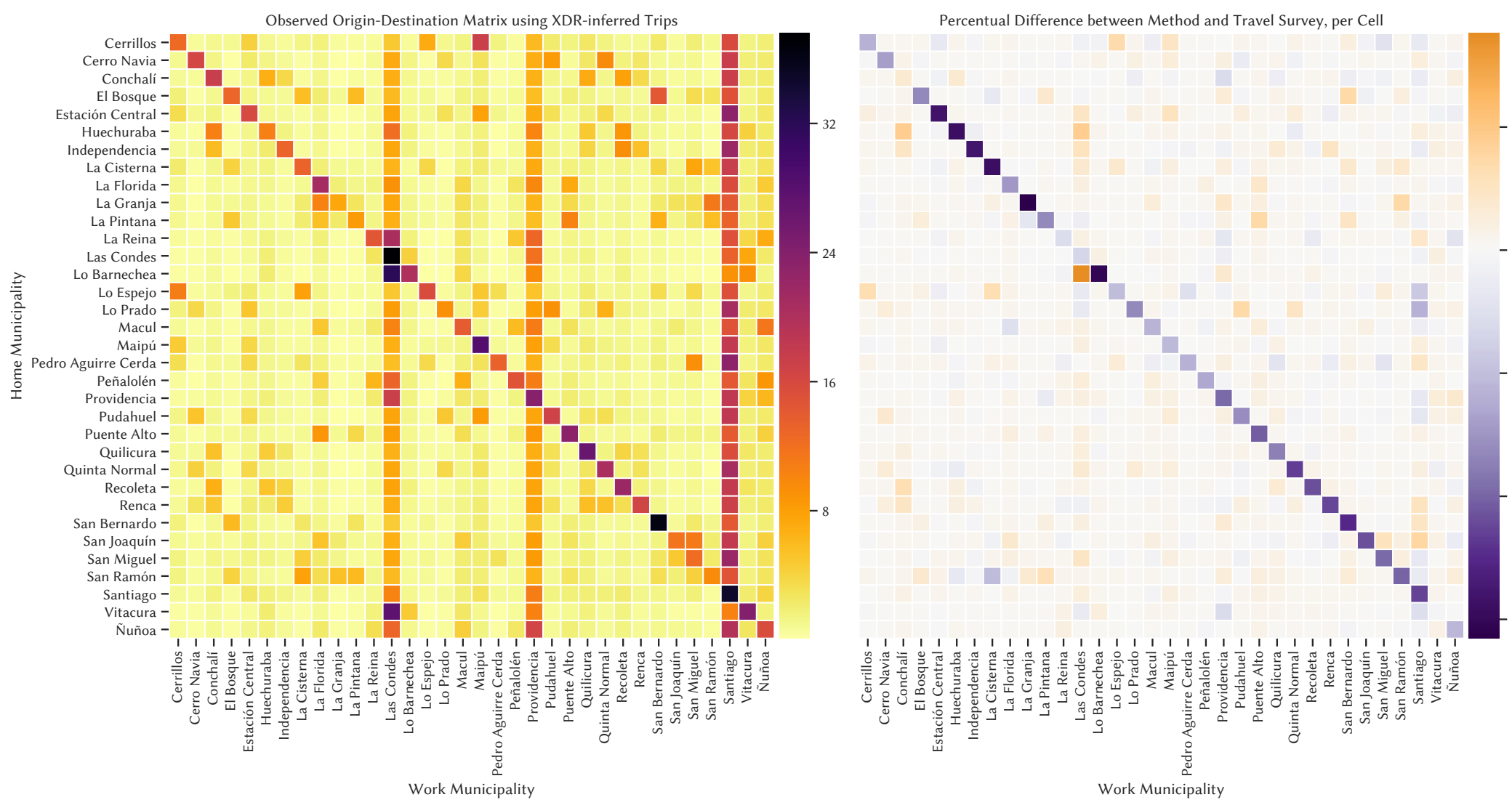


**Figure 3.** A codesigned visualization to compare two OD matrices. Left: a full OD matrix (row-normalized) inferred from mobile phone data. Right: differences between the full OD matrix from mobile phone data and the OD matrix from a travel survey.

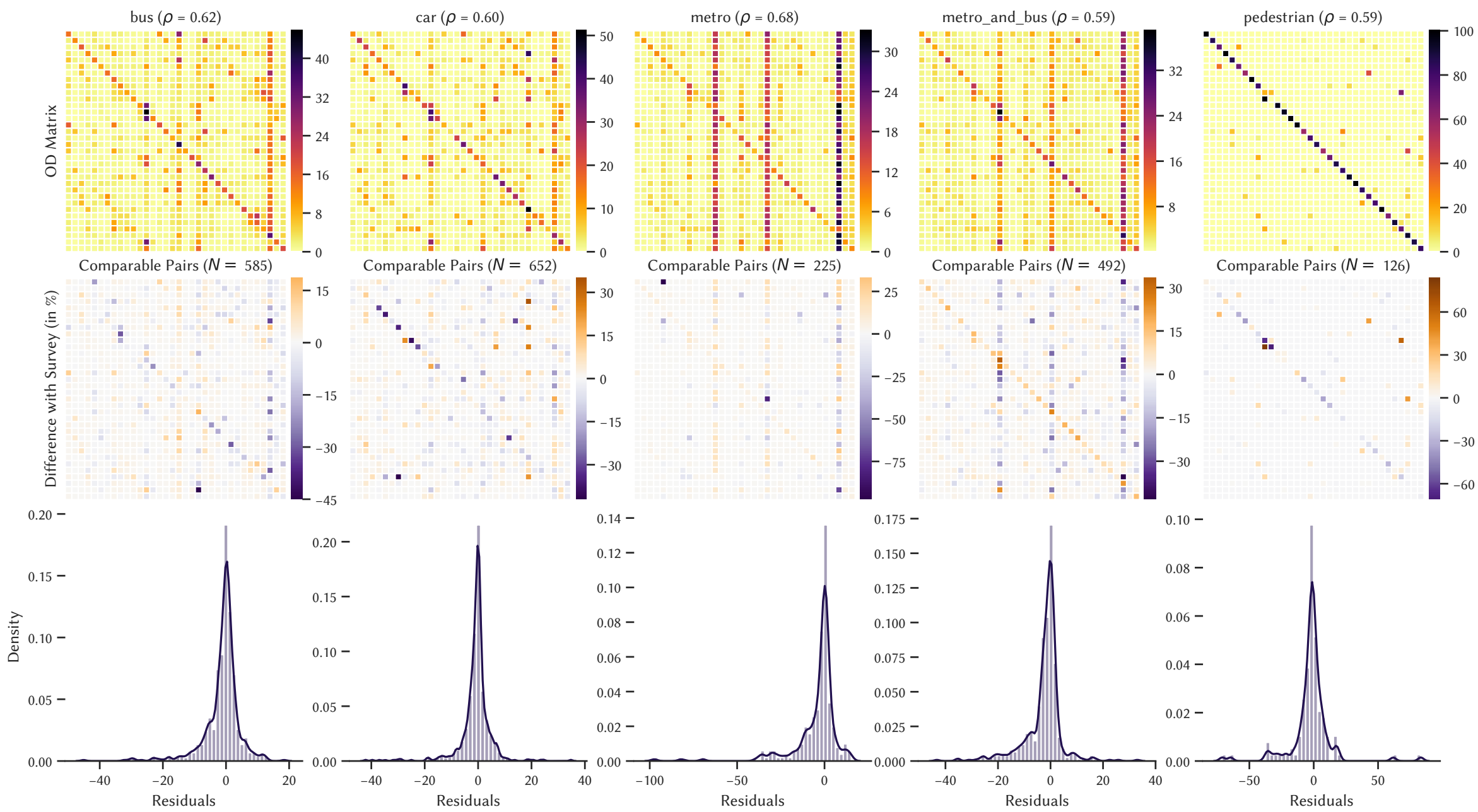


**Figure 4.** A codesigned visualization. The top row contains OD matrices per mode of transportation, inferred from mobile phone data. The middle row contains the difference between the previous matrices and those from the travel survey. The bottom row contains the distribution of these differences.

In summary, at the end of this stage both parties were satisfied with the results. Up to this point, the technical results of the project were published in a data science journal [17]. An academic research project may end there, but in our case, the project was applied and one of its goals was to be adopted in actual planning. We move in this direction in the next stages.

### *3.3. Interactive and Collaborative Evaluation*

From the interviews in the previous stage, we built a corpus of questions that were relevant for domain experts. However, these questions were focused on validating the inner workings on the proposed solution rather than on applications of it. Thus, in this stage, we aimed at identifying and performing actual tasks or application questions by potential users of a system that implemented our model. To do so, we designed and executed an interactive analysis session with domain experts to observe and learn from their usage of a proof-of-concept implementation of a visual dashboard [36].

Four potential end-users attended the interactive session: two policy makers from SECTRA and two practitioners from other institutions, one from a research center and one from the subway operator. It was held in a collaborative environment with large displays during two hours (see Figure 5 for the whole setup).

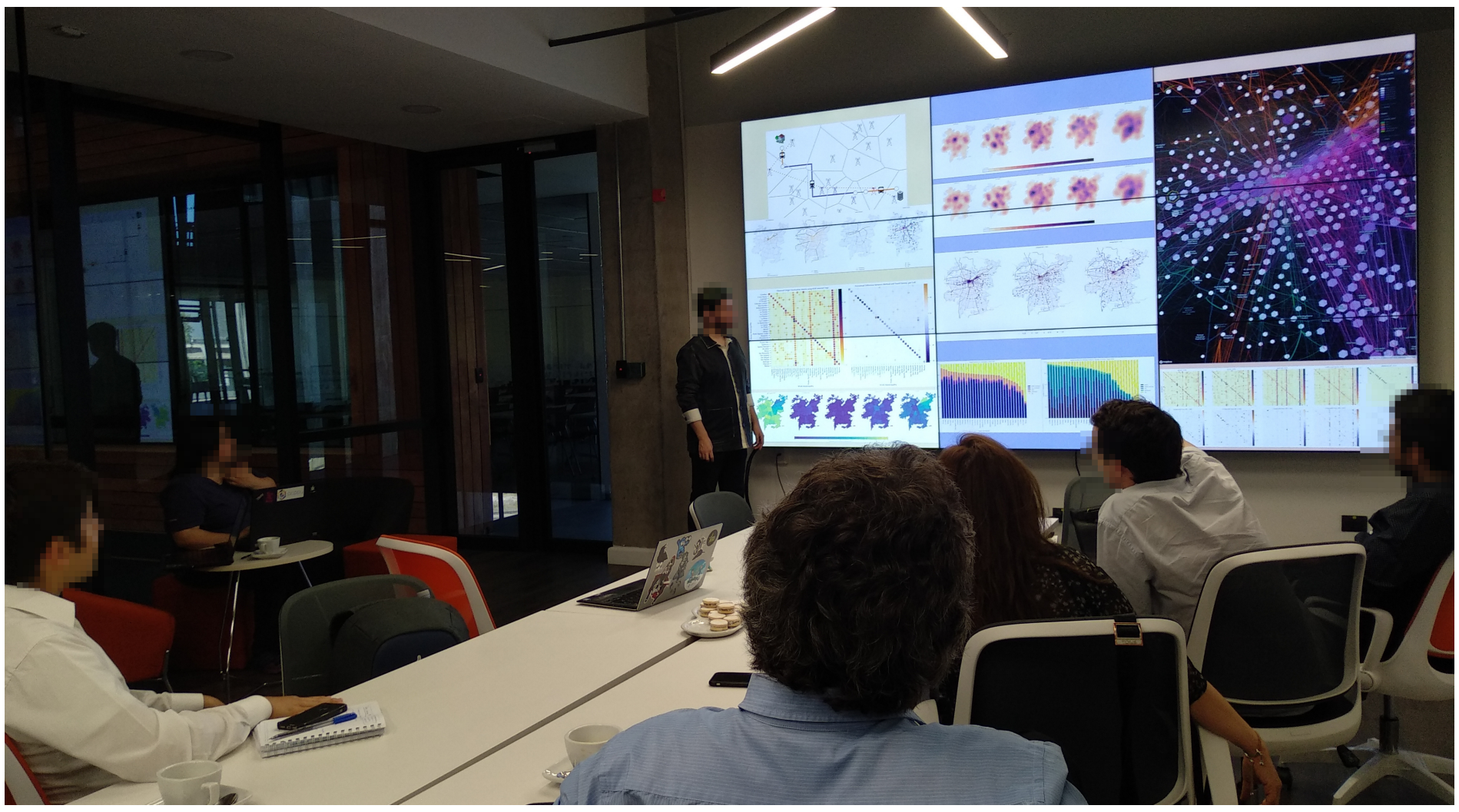

**Figure 5.** Interactive evaluation of the proposed model with domain experts in a collaborative environment.

The large display was composed of a 3 × 3 array of Samsung UD55E-B video wall, using the SAGE2 platform [45]. The results from the previous stages were visualized in two ways. Firstly, using the codesigned visualizations in real time on the large display through a Jupyter lab extension for SAGE2 [46]. Secondly, through an interactive visualization implemented in the geo-visualization framework Kepler.gl [47] which allowed an appealing and highly dynamic representation of the data (see Figure 6). The system allowed one to see OD flows within the city and filter them according to the mode of transportation and place of origin, as well as distributions of mode of transportation usage in a hexagonal grid laid out over the city. The room was arranged as a round table in which all participants could see each other and interact in a comfortable way. Participants were able to connect their mobile devices to the screen and use an interactive pointer within the system, as a way to enable single and collaboratively built queries and questions.

After the session, participants filled out a questionnaire regarding their experience. However, the most fruitful information was extracted from the conversation between experts during the session. From their perspective, this kind of model could be valuable; however, they questioned how the model outcomes could be integrated in their own pipeline of tools, both in terms of operational and methodological aspects. On the one hand, they needed more estimations of the data and model bias and errors, which were still not provided by the visualization nor the model. This may

be due to the expectations that data science solutions create on the domain experts, for instance, they requested information about the first and last stages of trips. The model only identifies whether people use multimodal transportation, not necessarily the order of modes in their commuting trips; thus, these features are not available. On the other hand, they declared the importance of including socio-demographic information about the people from whom this data was generated and to test the results with more cities to see how the model behaved in different conditions. However, we were not able to provide a temporary solution due to the anonymized nature of the data and the unavailability of data from other cities.

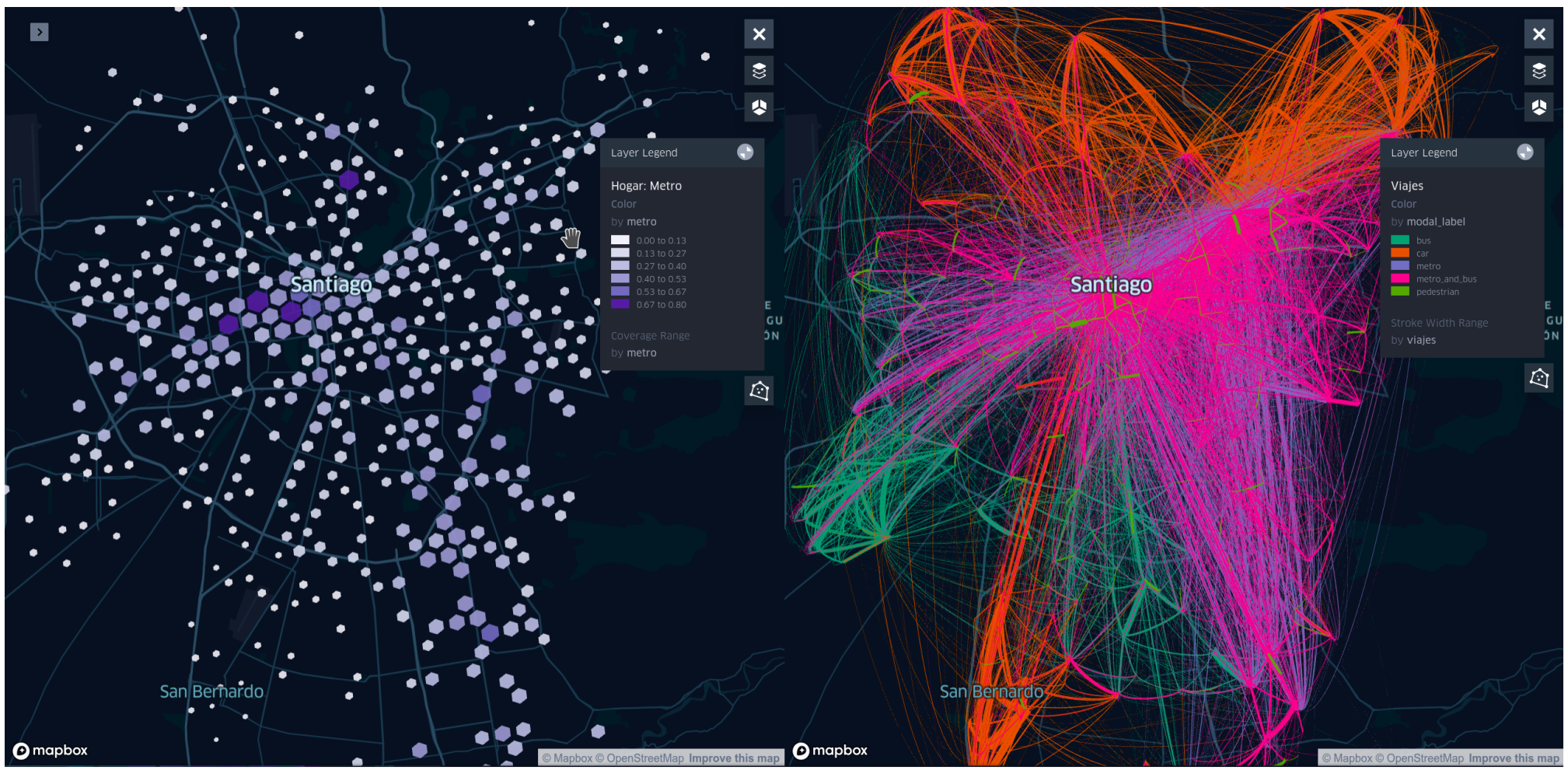


**Figure 6.** Interactive visualization used by the domain experts in the interactive session. This view shows a coordinated view with the spatial distribution of metro users (left) and the entirety of flows estimated by the model, colored according to primary transport mode per flow.

None of these insights were evident on the previous stages of the project, implying that the criteria to adopt the proposed solution is different than when designing it. We believe that this is due to the possibility of a *domain-directed* discussion enabled by the collaborative environment, instead of a *tool-directed* discussion (as in previous stages) in traditional environments or settings.

The interactive evaluation helped us to shape the next iteration of the system. We planned to perform a second session; however, the COVID-19 contingency made the session impossible to perform. These unexpected events are also part of the project pipeline, and it is important to be able to react to them. Fortunately, given our good relationships with domain experts, they have been available to discuss and test new iterations of the proposed solution, although in a more informal manner. The interactive session and its follow-up activities helped us to shape the next stage of the project, the consolidation of the generated knowledge.

### 3.4. Knowledge Consolidation

Finally, in this stage we consolidated the knowledge generated and obtained during the previous parts of the project. This consolidation has to deal with several adoption concerns that may be raised by stakeholders, which include domain experts, the data provider, and policy makers. Here we describe the several concerns that we identified and how we approached them. We do so by separating them into three aspects: *system integration*, *intellectual property*, and *technology transfer*.

The *system integration* aspect refers to a key insight we obtained during the interactive evaluation. In the previous stages, the main adoption concerns for transportation planners were, in addition to performance, interpretability and transparency. Here, *interoperability* surfaced as an equal or even more important quality of the system. Interoperability is seen by the domain experts as a way of updating and calibrating already existing models and systems, which the domain experts trust (and will not stop using in the short term), and generating fine-grained input data for their systems. Indeed,

transportation experts are looking for ways to modernize their tools and methodologies instead of replacing them. Hence, an adoption concern is not whether the model works or not (it does), it is its feasibility of integration with existing tools used by planners.

Regarding the *intellectual property* aspect, one concern by the data provider was how this system could be made sustainable in time. This requires, for instance, the protection of any invention generated during the project through a patent. This raises concerns also for the research center, because the IP rights may be shared or not between both actors. In our case, both the research center and the telecommunication company applied for a patent together in Chile [48]. Furthermore, in later interviews with domain experts, we learned that having a patent improves their trust in the project. This is due to planning agencies being constantly offered with digital services based on Big Data platforms, which has decreased their confidence due to overpromising marketing.

With respect to *technology transfer*, as a research center we do not develop production-ready data products, and the telecommunication company itself does not develop such solutions either. This is a potential adoption concern, as there are risks of not being able to provide a usable data product. To avoid this in our project, both actors coordinated efforts in producing a prototype implementation that can be used as a base for a production-ready product. To demonstrate the value for the community, we have open-sourced two of the key elements of our project. On the one hand, we produced a efficient implementation of the algorithm employed in the model, TS-NMF [49], which works with sparse matrices and thus is able to process huge sets of digital traces. On the other hand, taking into account the observations of how domain experts interacted with the proof-of-concept implementation from the previous stage, we implemented two different visualizations in the system, one novel designed for this problem and one previously known. The first one is a design based on domain requirements, which focuses on the travel demand that a place *generates* or *attracts*. This design is called ModalCell (see Figure 7) and was presented at a relevant visualization conference [50]. The second one is a functional and attractive flow visualization that displays OD pairs (flows) interactively (see Figure 8). It uses an already available library named *flowmap.gl* [51]. Both are integrated into a dashboard that is ready to load flow data with distributions of mode of transportation, as generated by our model [52].

The lessons we discussed in this section are summarized in Table 1. We have separated the insights in two categories: learned during the execution and learned after a posterior stage was executed—for instance, interoperation with other systems is necessary. We learned this in the interactive evaluation stage; however, future projects should consider it in the algorithm-design stage or in the data-driven evaluation stage, depending on the project. Although the methodology is defined in a sequential manner, the insights we obtained suggest that the sequence of steps is not necessarily linear, as every stage has consequences that may propagate to the others in a nested way [53].

After following the methodology, the end-result was a *solution* for a transportation problem based on data science. The methodology and its execution were designed having adoption of the solution as one key objective. We can comment that, indeed, the solution was adopted—it is being used at the time of writing this paper to infer the OD matrix and its corresponding modal partition in the conurbation of Rancagua–Machalí, in the south of Santiago. This conurbation is much smaller than Santiago, with around 300K inhabitants, and is being surveyed with a travel survey at the same time, which will allow the generation of an effective ground truth dataset to evaluate the system and serve as a pilot for the entire collaborative effort between these three institutions.

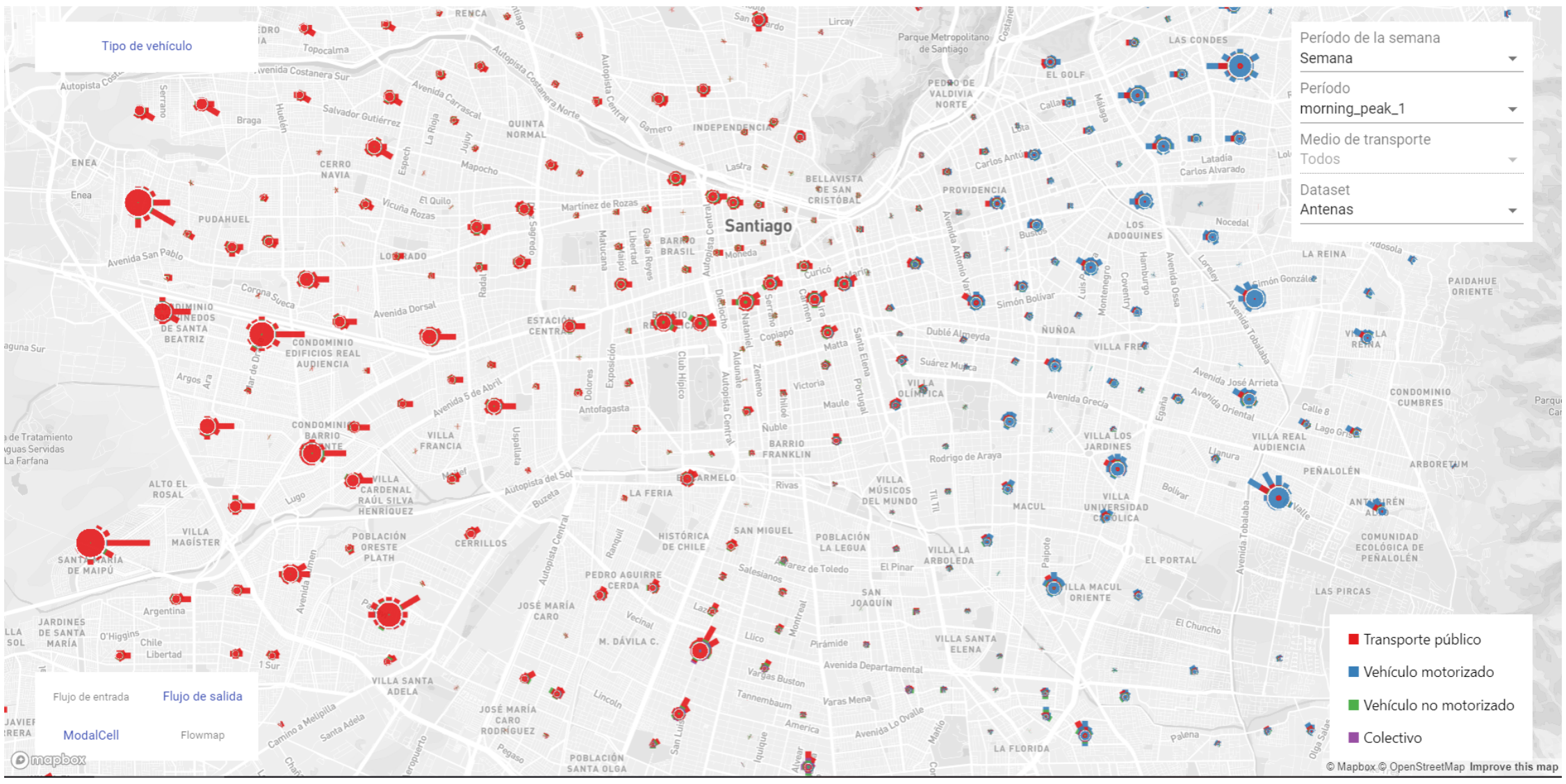


**Figure 7.** Implementation of the ModalCell visualization [50] in the final version of the project. Each glyph represents the origin of flows, with bars encoding their direction and magnitude, and bar colors indicating the prominence of each mode of transportation.

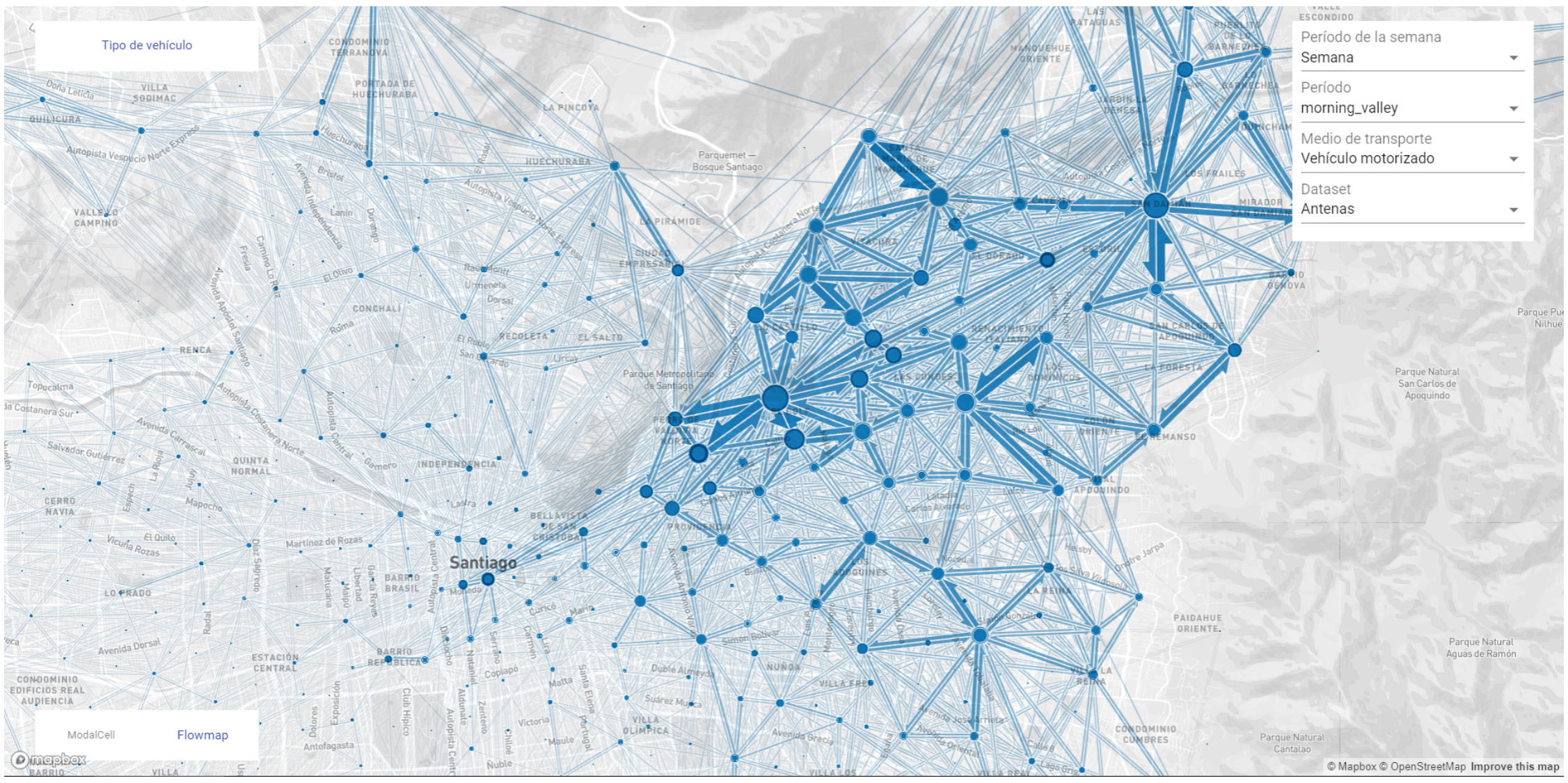


**Figure 8.** Flow visualization (trips between origin and destination areas) using the *flowmap.gl* toolkit.

**Table 1.** Summary of lessons learned during the project.

| Stage | Lessons Learned during Execution | Lessons Learned from Other Stages |
|---|---|---|
| Algorithm-Design | Semisupervised methods help to integrate domain knowledge into models. Find how underlying methods in model definition map to common operations in domain practice (e.g., optimization methods). | Ensure that algorithm output can be integrated into existing domain tools. Examples include file-formats or plugin-based implementations. Be careful with hidden/implicit expectations about model outcomes. |
| Data-Driven Evaluation | Interpretability has many dimensions, in transportation problems it may refer to analyzing the model errors. Codesigned visualization is a powerful device to assess model behavior. | Demographic or population representativeness of source data (digital traces) and model outcomes should be evaluated as soon as possible in the project pipeline. |
| Interactive Evaluation | Collaborative environments help to surface interactions between domain experts that are less likely to be visible in traditional environments. | In addition to prototype-implementation with modern technology, evaluate proof-of-concept interoperability with existing tools. |
| Knowledge Consolidation | Agreement about which components of the proposed solution will be subject to intellectual property rights is needed. Furthermore, patents increase trust in the system. | — |

## 4. Conclusions

In this work we defined a methodology to solve transportation problems using data science, through a collaborative effort. This methodology was applied in the design and implementation of a model to work on a current problem, the inference of the mode of transportation share in a city of 8 million inhabitants. It allowed us to identify key qualities in the model and its depiction that were needed to generate actionable insights from the system, as well as to identify application and adoption concerns that need to be addressed before a concrete usage of it. Furthermore, we believe that the trust built in the collaboration with transportation authorities and operators played a role in starting the first official usage of mobile phone data in the understanding of transportation patterns in Chile.

We described the lessons learned at each stage of the project. For instance, *interpretability* was explicitly important in the first stages of the project, whereas *interoperability* surfaced as important when concluding it, even though it is a critical quality needed for adoption. The importance of interoperability opens paths not only to data integration but also paradigm integration. Transportation planners make use of transportation simulation software, which could benefit from incorporating the results of data-driven projects, in tasks such as model calibration and demand prediction. As a benefit for the transportation planning software, it would allow practitioners to define how to incorporate situations and data structures that do not comply with the assumptions inherent in the simulation (e.g., the disruption of new modes of transportation). Such simulations may be able to feed important domain knowledge into a data-driven system, creating a positive feedback loop that complements the communication channels devised in this work. Thus, having this last trait in mind when starting the project would impact how the algorithms are designed and evaluated and also how they are made available to the stakeholders and the community. A limitation in our methodology is the lack of a proper definition of when and where to perform this integration. This is left for future work.

One of the limitations of this paper is the preliminary aspect of the methodology. We have applied it to a specific problem, with positive results, but it still remains a challenge to see how it would behave in other projects that are not so tied to its definition. Moreover, it is not clear whether the problem could have been solved effectively using a different methodology. However, in contrast with common user studies, the number of available end-users for this type of task is rather limited, thus, a set of guidelines that help to build trust and to improve the qualities of the model is a valuable asset.

As future work, we aim to continue studying how machine learning, visualization, and user research could impact the adoption of nontraditional data sources for urban planning and policy making. Doing so is critical in times of rapid changes in behavioral patterns but also in times of crisis. Cost-effective and ready-to-apply solutions may help in improving accessibility and sustainability of transport networks in cities.

**Author Contributions:** Conceptualization, E.G.-G. and V.P.-A.; methodology, E.G.-G.; software, E.G.-G.; validation, E.G.-G. and V.P.-A.; resources, E.G.-G. and L.B.; data curation, E.G.-G. and L.B.; writing–original draft preparation, E.G.-G. and V.P.-A.; writing–review and editing, E.G.-G., V.P.-A. and L.B.; funding acquisition, E.G.-G. and L.B. All authors have read and agreed to the published version of the manuscript.

**Funding:** E.G-G. was partially funded by CONICYT Fondecyt de Iniciación #11180913.

**Acknowledgments:** We wish to thank the Atelier Team at Inria Chile (http://inria.cl/) for help in organizing and holding the interactive session, particularly to Celeste Bertin for her technical support. We thank Sebastián Aedo and Víctor Navarro for their help implementing software. We also thank Francisca Arévalo for help in organizing and designing the pilot workshops and Karina Flores for interviewing experts in the last stages of the research. Finally, we thank Viviana Muñoz from the Secretary of Transportation for helping us to build bridges between disciplines.

**Conflicts of Interest:** The authors declare no conflict of interest. The funders had no role in the design of the study; in the collection, analyses, or interpretation of data; in the writing of the manuscript, or in the decision to publish the results.

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
