# Peer review of "Adoption-Driven Data Science for Transportation Planning: Methodology, Case Study, and Lessons Learned"

_sustainability, doi:10.3390/su12156001_

Round 1

Reviewer 1 Report

The manuscript explores the connection between the traditional field of transportation planning and the emerging field of data science, and proposes a 4-step method to guide the collaborative development of data-driven solutions. The method is illustrated on a case study of trips inferred from mobile phone data.

The study is of very high quality in every aspect: It clearly states the issues, the methodology, and adequately cites the literature. Figures are also clear and support the text well. The research question is highly topical and original. The description of experiences, and of the adapting project pipeline, of solving a data/transport issue in a collaborative environment are refreshing and important. State of the art technologies, such as kepler.gl are being used. The research question is highly relevant for sustainable urban transport development and policy making, therefore the manuscript fits perfectly in the journal. It would be of high interest for its readers.

Author Response

We thank the reviewer for the encouraging feedback.

Reviewer 2 Report

Page 1, line 14: “Cities are becoming more complex”. Please define “complexity” for the purpose of your research.

Page 1, lines 19, 20: “Arguably, even though both disciplines work on similar problems, they use a different domain language, resort to different data sources and methodologies, and have different priorities.”. I am not sure if I can agree with this statement. “data science” provides the tools to “transportation” to identify, understand, evaluate and solve problems. They do not just simply work on similar problems. They complement one another.

Page 1, lines 20,21: “….Due to this gap between both fields,…”. You use either the term “disciplines” or “fields”. I think that you must keep “disciplines”.

Page 2, line 34: “………..in a sustainable way for all actors involved….”, Page 2, line 42: “….outcomes are adopted by relevant stakeholders…..”. You use either the term “actor” or “stakeholder” for the same purpose. Please keep “stakeholder”.

Page 13, Table 1. Summary of lessons learned during the project. The heading of Table 1 should appear above the Table.

Page 2, lines 43,44: “….that puts data science and transportation to work collaboratively to solve transportation problems….”. It is not only a matter of “solving problems. It is also a matter of identify, understand, and evaluate. You actually refer to this by saying “....This methodology is designed to be applied after a transport planning problem has been identified….” (page 2, line 71). The same applies for “evaluation”, (Figure 1). You must clear things at the beginning of the paper (abstract).

Section 1, Introduction is rather weak in terms of literature review (11 references). The topic of the paper is wide enough, so please try to enhance Section 1.

Page 2, line 59,60: “….Our main goal is to define a framework to guide the collaborative development of data-driven solutions for problems in transport planning….”. The title and abstract refer to “transportation” in general. You should refer specifically to “transportation planning".

Taking into account Figure 1, I think that “transportation planning” should be included in the title of the paper.

Page 5, lines 183-185: “ ….It is a complex scenario, as they may have to look into at many more dimensions of the problem that the technical actors involved are able to see….”. Please briefly explain the reason(s) why the technical actors cannot (always?) see all the dimensions of the problem.

Page 6, line 220: (e.g.,Waze) Please provide some more details about Waze (i.e., navigation and live traffic) for the readers who are not familiar with Waze.

Figure 5. Interactive evaluation of the proposed model with domain experts in a collaborative environment. Maybe you must hide the faces of the people appearing in Figure 1 (for privacy protection reasons).

Page 12, line 402: “……and the telecommunication company. itself doesn’t develop such solutions either…”. Please delete the “dot” between “company” and “itself”.

Section 4. Conclusions: The traditional method to see the flows between an origin and a destination and to evaluate various scenarios makes use of a transportation simulation software. Data input usually comes from questionnaire surveys (i.e., home-based) and the use of mobile phones as well. Calibration of the model is based on the comparison of actual traffic counts (observed values) with the respective model output. I will welcome a paragraph in the conclusions dedicated to the advantages and disadvantages of your research when it will be compared to the use of a transportation simulation software.

Author Response

We are thankful for the constructive feedback provided by Reviewer 2. The paper is stronger now thanks to the suggestions. Here we provide a point by point summary of changes in the paper. To simplify revision, here we have copied the main text related to the changes, highlighting the changes with a different color.

Page 1, line 14: “Cities are becoming more complex”. Please define “complexity” for the purpose of your research.

Fixed. We have changed the first lines of the paper to the following:

Cities are becoming increasingly complex, growing larger due to urbanization processes~\cite{ritchie2018urbanization}, and with increasing layers of interaction between their populations and their urban infrastructure~\cite{batty2012building}. The discipline of transportation is particularly affected by these issues, not only due to city growth, but also to the arrival of new technologies and to changes in urban behavior.

Page 1, lines 19, 20: “Arguably, even though both disciplines work on similar problems, they use a different domain language, resort to different data sources and methodologies, and have different priorities.”. I am not sure if I can agree with this statement. “data science” provides the tools to “transportation” to identify, understand, evaluate and solve problems. They do not just simply work on similar problems. They complement one another.

We value this comment. We have updated the mentioned text with the following:

Arguably, both disciplines complement each other: data science may provide tools to transportation to identify, understand, evaluate, and solve problems; transportation may provide important domain problems to be solved with data-driven approaches. However, both use a different domain language, resort to different data sources and methodologies, and have different priorities.

Page 1, lines 20,21: “….Due to this gap between both fields,…”. You use either the term “disciplines” or “fields”. I think that you must keep “disciplines”.

Fixed. We replaced occurrences of field(s) with discipline(s).

Page 2, line 34: “………..in a sustainable way for all actors involved….”, Page 2, line 42: “….outcomes are adopted by relevant stakeholders…..”. You use either the term “actor” or “stakeholder” for the same purpose. Please keep “stakeholder”.

Fixed. We replaced most occurrences of actor(s) with stakeholder(s).

Page 13, Table 1. Summary of lessons learned during the project. The heading of Table 1 should appear above the Table.

Fixed.

Page 2, lines 43,44: “….that puts data science and transportation to work collaboratively to solve transportation problems….”. It is not only a matter of “solving problems. It is also a matter of identify, understand, and evaluate. You actually refer to this by saying “....This methodology is designed to be applied after a transport planning problem has been identified….” (page 2, line 71). The same applies for “evaluation”, (Figure 1). You must clear things at the beginning of the paper (abstract).

We thank the reviewer for this comment. We have updated the introduction and the abstract to improve the text in the suggested direction:

Here we propose to bridge the gap toward adoption through a four-step methodology that puts data science and transportation to work collaboratively to identify, understand, solve, and evaluate solutions to transportation planning problems. Some of these problems are not new[...].

In the abstract:

In this context, this paper proposes a methodology toward bridging two disciplines, data science and transportation, to identify, understand, and solve transportation planning problems with data-driven solutions that are suitable for adoption by urban planners and policy makers. The methodology is defined by four steps where people from both disciplines go from algorithm and model definition to the building of a potentially adoptable solution with evaluated outputs.

Section 1, Introduction is rather weak in terms of literature review (11 references). The topic of the paper is wide enough, so please try to enhance Section 1.

Agreed. We have expanded the following paragraphs in section 1:

There are methodologies that put values and well-being as ultimate aims of a data science project~\cite{zhu2018value}, and while we agree that data science projects should strive for improving quality of life through social impact, this can be obtained only if its outcomes are adopted by relevant stakeholders. Indeed, there are extensive Big Data projects documented in the literature that cover transportation demand analysis from digital traces~\cite{toole2015path}.To the extent of our knowledge, it is not reported how or when these systems were put into operation in case they were adopted in real planning scenarios.

Most of the work in collaborative efforts between data science with other disciplines has focused on intermediary steps in the path toward adoption, but not adoption itself. For instance, in the definition of liaison roles that improve the technical quality of a project~\cite{simon2015bridging}, the definition of methods to synthesize the results of collaborative activities between stakeholders~\cite{robinson2008collaborative}, and the definition of analysis approaches where domain experts have guided interactions with analytical systems to maximize their usefulness~\cite{wood2014moving}. Here we propose to bridge the gap toward adoption through a four-step methodology that puts data science and transportation to work collaboratively to identify, understand, solve, and evaluate solutions to transportation planning problems. Some of these problems are not new[...].

Page 2, line 59,60: “….Our main goal is to define a framework to guide the collaborative development of data-driven solutions for problems in transport planning….”. The title and abstract refer to “transportation” in general. You should refer specifically to “transportation planning".

Taking into account Figure 1, I think that “transportation planning” should be included in the title of the paper.

Agreed. Now the title is Adoption-Driven Data Science for Transportation Planning: Methodology, Case Study, and Lessons Learned.

Page 5, lines 183-185: “ ….It is a complex scenario, as they may have to look into at many more dimensions of the problem that the technical actors involved are able to see….”. Please briefly explain the reason(s) why the technical actors cannot (always?) see all the dimensions of the problem.

Agreed. To make it clearer we rephrase it in this way:

In this stage stakeholders need to assess what is needed for the proposed solution to be adopted in a real planning scenario. We advance in that direction by jointly identifying concerns relevant for the adoption of the project. From their position, stakeholders look into dimensions that are orthogonal to the technical aspects covered in the previous stages. These include: definition of intellectual ownership and exploitation agreements with the data provider; the cost of accessing, storing, and updating the data; the compliance with privacy regulations; and the limits of population representativeness in the available digital traces, among others.

The key change is the use of the word orthogonal, which implies that these aspects are not strictly needed for the technical solution to be achieved.

Page 6, line 220: (e.g.,Waze) Please provide some more details about Waze (i.e., navigation and live traffic) for the readers who are not familiar with Waze.

Fixed: (\textit{e.g.}, Waze, an application that provides navigation and live traffic)

Figure 5. Interactive evaluation of the proposed model with domain experts in a collaborative environment. Maybe you must hide the faces of the people appearing in Figure 1 (for privacy protection reasons).

Fixed: visible faces were pixelated.

Page 12, line 402: “……and the telecommunication company. itself doesn’t develop such solutions either…”. Please delete the “dot” between “company” and “itself”.

Fixed.

Section 4. Conclusions: The traditional method to see the flows between an origin and a destination and to evaluate various scenarios makes use of a transportation simulation software. Data input usually comes from questionnaire surveys (i.e., home-based) and the use of mobile phones as well. Calibration of the model is based on the comparison of actual traffic counts (observed values) with the respective model output. I will welcome a paragraph in the conclusions dedicated to the advantages and disadvantages of your research when it will be compared to the use of a transportation simulation software.

We thank the reviewer for this comment. Indeed, it is something important that needs to be addressed. To do so, this text was added to a paragraph in conclusions:

We described the lessons learned at each stage of the project. For instance, \emph{interpretability} was explicitly important in the first stages of the project, whereas \emph{interoperability} surfaced as important when concluding it, even though it is a critical quality needed for adoption. The importance of interoperability opens paths not only to data integration but also paradigm integration. Transportation planners make use of transportation simulation software, which could benefit from incorporating the results of data-driven projects, in tasks such as model calibration and demand prediction. As a benefit for the transportation planning software, it would allow practitioners to define how to incorporate situations and data structures that do not comply with the assumptions inherent in the simulation (e.g., the disruption of new modes of transportation). Such simulations may be able to feed important domain knowledge into a data-driven system, creating a positive feedback loop that complements the communication channels devised in this work. Thus, having this last trait in mind when starting the project would impact how the algorithms are designed and evaluated, and also, how they are made available to the stakeholders and the community. A limitation in our methodology is the lack of a proper definition of when and where to perform this integration. This is left for future work.